# Vitality of Urban Parks and Its Influencing Factors from the Perspective of Recreational Service Supply, Demand, and Spatial Links

**DOI:** 10.3390/ijerph17051615

**Published:** 2020-03-02

**Authors:** Jieyuan Zhu, Huiting Lu, Tianchen Zheng, Yuejing Rong, Chenxing Wang, Wen Zhang, Yan Yan, Lina Tang

**Affiliations:** 1Key Laboratory of Urban Environment and Health, Institute of Urban Environment, Chinese Academy of Sciences, Xiamen 361021, China; jyzhu@iue.ac.cn (J.Z.); lntang@iue.ac.cn (L.T.); 2University of Chinese Academy of Sciences, Beijing 100049, China; htlu_st@rcees.ac.cn (H.L.); tczheng_st@rcees.ac.cn (T.Z.); 3State Key Laboratory of Urban and Regional Ecology, Research Centre for Eco-Environmental Sciences, Chinese Academy of Sciences, Beijing 100085, China; rongyuejing@126.com (Y.R.); cxwang@rcees.ac.cn (C.W.); zwzzzz@whut.edu.cn (W.Z.); 4School of Resource and Environmental Engineering, Wuhan University of Technology, Wuhan 430070, China

**Keywords:** park vitality, recreational services, volunteered check-in data, POI-based urban function mix, Beijing

## Abstract

Urban parks provide multiple non-material benefits to human health and well-being; measuring these “intangible” benefits mainly co-produced by the spatial interactivity between dwellers and urban parks is vital for urban green space management. This paper introduced “vitality” to measure the intangible benefits of urban parks and constructed a straightforward and spatially explicit approach to assess the park vitality based on visiting intensity and recreational satisfaction rate. Freely available data of check-in comments on parks, points-of-interest (POIs), and other multi-source data from Beijing were used to assess the urban park vitality and explore the factors influencing it from the perspectives of recreational service supply, demand, and spatial linking characteristics. We found that the urban park vitalities decreased along the urban–rural gradient. The presence of water and facility density in the parks have significant positive impacts on park vitality, and high population density nearby was a positive factor. Moreover, the external higher levels of the POI-based urban function mix and density, as well as developed public transportation, were strongly associated with greater park vitality. Our research proposed a feasible and effective method to assess the park vitality, and the findings from this study have significant implications for optimizing the spatial configuration of urban parks.

## 1. Introduction

Human health and well-being are linked to the natural environment in a myriad of ways [1]. With rapid urbanization and the increasing urban population, the demand for contact with nature and high-quality life for urban residents is also increasing. Urban green space provides a number of cultural ecosystem services that include spiritual enrichment, cognitive development, reflection, sense of place, and recreational experiences [2], which play an important role in improving the physical and psychological health of residents, maintaining good social relations, and improving overall quality of life [1,3,4,5]. These non-material benefits provided by urban green spaces have intrigued many researchers; however, due to their “intangibility” and “subjectivity” [6,7], they are commonly assessed by self-reporting methods [8,9]. Stålhammar (2017) used interpretative methods to explore how individuals perceive and value their recreational experiences in natural environments in their own words [8]; the benefits were perceived through naturalness and experiences of authenticity, healing, beauty, magic, and movement. These non-material benefits are challenging to be accurately quantified and operationalized by using biophysical indicators [10].

Urban parks, particularly for urban residents, are the most important part of urban green space, and their main benefits or values are driven by the residents’ demand for recreational services [11,12]. Recreational services cover a wide variety of activities, such as walking, jogging, running, picnicking, and aesthetic experiences in nearby urban parks. Within the range of cultural ecosystem services relevant to urban parks, recreational ecosystem services are particularly important and widely acknowledged for securing mental and physical wellbeing [13,14]. Since recreational services are “user movement related” [15], the “intangible” benefits mainly arise from the actual contact and spatial interactivity between residents and accessible nature-based areas (i.e., urban parks) (Figure 1) [16,17,18]. The natural and semi-natural landscapes are the service providing areas (SPAs) and produce recreation potential or opportunities given by their biophysical attributes [6,19]. The service benefiting areas (SBAs) are the areas where the recreationists contact with SPAs to benefit from the recreational services [19]. The recreation potential differs from the recreation benefit (the realized services) that directly contribute to human well-being [6]. That is, the recreation benefit is a result of the users’ recreational experience on-site if the user can reach the SPAs [19]. If people are unable to access the SPAs, the recreation potential cannot become a service, because there is no actual use of recreational services [18]. The periphery of the park is the service connecting areas (SCAs) between the SPAs and the areas in which beneficiaries are located (i.e., residential area). Therefore, the benefits of recreational services provided by urban parks are affected by recreational supply and demand factors as well as the spatial links in SCAs [20,21]. The question of how to measure and map the intangible benefits co-produced mainly by recreationists and urban parks and exploring what factors affect these benefits are vital for urban green space planning and management as well as human well-being improvement [4,7].

A growing number of studies that address these benefits have focused on monetizing the urban green spaces and their spatial differences using hedonic pricing, land rent methods, and survey methods based on willingness to pay [22,23]. However, the benefits estimated by the monetizing method, such as the assigned land price, do not fully capture people’s demand for or utilization of cultural and recreational services. Meanwhile, willingness to pay for nature is more closely related to individual socioeconomic status and subjective cognition of green protection rather than a high attachment to or use of urban parks [24]. On the other hand, from the perspective of improving human well-being [2], the benefits of urban parks are captured by the satisfaction level of the residents’ demand for recreational service quantity and quality. Human well-being represents how well human needs are met or the extent to which individuals or groups perceive satisfaction across multi-dimensional life domains [25]. Thus, the level of satisfaction of recreational demands or experiences is more conducive to revealing the benefits of using urban parks to human well-being than the virtual price. Up until now, spatially specific and robust assessment of the intangible benefits of urban parks that can offer a clear feedback mechanism to drive improved land management and intuitively reflect the contribution of urban parks to human well-being has been elusive.

To address the aforementioned issues, this study introduces “vitality” to measure the intangible benefits of urban parks and constructs a straightforward and spatially explicit approach of assessing the park vitality, then explores the factors influencing park vitality from the perspectives of recreational service supply, demand, and spatial linking characteristics. We believe that this study could contribute to sustainable urban research aimed at improving human well-being and provide implications for urban green space management for urban planners and decision-makers.

## 2. Introducing Vitality to Measure the “Intangible” Benefits of Parks

Considering that the intangible benefits are mainly co-produced by the accessible interactivity between residents and urban parks, this paper introduces vitality to measure the benefits instead of a virtual monetized price. The concept of vitality is closely related to the human activity intensity in urban public spaces [26,27]. Jacobs (1961) claimed that people’s activities and the diversity of their lives have bred urban vitality [26], which usually refers to the capacity of a place to induce lively social and economic activities as well as cultural exchanges [28]. Similarly, we define park vitality as the liveliness that attracts urban residents to use the recreational services sustainably due to the richness of the park landscape, diversity of functions, and accessibility of the park. Besides, we propose that from the “recreational service produce–utilization–human well-being” cascade view [29], park vitality mainly depends on the users’ actual contact with parks, the utilization of different recreational activities, and the satisfaction degree of recreational experiences by individuals. Due to the recreation benefit resulting from the users’ recreational experience on-site [19], the level of recreation use, measured as visitors flow, is one of the possible proxies of the benefit delivered by the recreational services [6]. On the other hand, the subjective satisfaction of the recreational experience highly contributes to human psychological well-being [30]. In recreation research, satisfying recreational experiences depend on the design of natural and manmade elements, and on amenities meeting visitors’ preferences and demands [31]. For the park itself, the greater visiting intensity of the urban park and the higher degree to which the visitors’ recreational demands are met, the greater the park vitality will be, which means that the benefits of using recreational services contribute more to residents’ well-being. Therefore, we can construct the park vitality evaluation method based on two indicators: visiting intensity and recreational satisfaction rate, making the intangible benefits measurable.

The rapid advancement of location-based services and data (e.g., mobile phone signaling data, points-of-interest (POIs), social media check-in data), as well as their extensive permeation into human activities, provide support for exploring the vitality of urban public spaces in-depth [27,32]. The emergence of freely available social media data from different locations, as one of the most popular types of geo-tagged data in a city, provides new approaches for measuring the visits to green space or parks and identification of nature-based recreation preferences [33,34]. Schirpke (2018) used the online photo-sharing website Flickr to map outdoor recreation hot spots in the European Alps and surrounding areas [35]; Zhang (2018) used geo-tagged Weibo check-in data to identify and map visits to different types of parks in central Beijing, China [36]. In addition, related studies have confirmed that social media data can be a reliable proxy for outdoor recreation visiting frequencies, and the data are well-correlated with official visitation statistics [37,38].

On the other hand, figuring out what factors affect the vitality of urban parks is an important issue for sustainable urban development. At present, many studies have found that both physical attributes (such as park size, amenities, and maintenance), demographic characteristics (such as age, gender, and educational level), and accessibility were associated with park use [20,36,39]; however, few studies have examined the effects of the spatial linking characteristics of SCAs on the use or vitality of urban parks. The SCAs are a block scale area composed of various urban functional facilities and residents’ spatial activities, and the spatial links of SCAs mainly include two aspects: traffic accessibility and urban function characteristics. At the city level, urban functions are related to the use and functions of different urban spaces and include residential, productive, social, commuting, recreational, and administrational activities, and urban function can be conceptualized as activities taking place inside of cities [40]. Mixed and multifunctional uses usually refer to a combination of residential, commercial, cultural, institutional, and industrial uses, which have been identified as being able to promote urban vibrancy [26,41]. The intensity of development and diversity of urban functions may lead to differences in the population structures and social activities [42], which may affect the interaction between residents and nearby parks. Knowledge about the effects of the urban function characteristics affecting urban park vitality helps in predicting the extent and range of possible benefits from different configurations of urban parks at a city or urban block scale.

To illustrate and validate this approach, we quantified and mapped the urban park vitality in Beijing as an exemplary case and then explored the factors influencing park vitality by taking advantage of social media check-in data, POIs, and other multi-source datasets. Finally, we highlight the effects of urban function characteristics on park vitality and provide important insights into urban park management and planning.

## 3. Materials and Methods

### 3.1. Study Area

This study was conducted in Beijing, the capital city of China. Beijing has an administrative area of 16,808 km^2^. By the end of 2018, there were 21.54 million permanent residents in Beijing, with a population density of 1313 people per km^2^. According to the statistics from Beijing Landscaping Bureau, in 2018, the total area of urban green space in Beijing was 32,618.50 ha, with a per capita green space area of 16.30 m^2^. In this study, we focused on the area within the 6th ring road of Beijing (Figure 2), which is composed of an urban core area and part of a suburban area, with an area of approximately 2267 km^2^. There were 16.34 million permanent residents within the 6th ring, accounting for 75.94% of the whole population in Beijing, and the area had a huge recreational service demand for urban green space. We selected and mapped the urban parks mainly according to the park lists from the Beijing Municipal Bureau of Landscaping (http://yllhj.beijing.gov.cn/) and park distributions provided by Gaode Map (https://lbs.amap.com/) in ArcGIS 10.3. Since Beijing is the political, historical, and cultural center of China, there are many types of urban parks, and some famous historical relic parks attract a large number of non-local tourists every year. The non-local visitors might be more likely to “check-in” the famous historical relic parks, which may increase potential differences among parks. To better analyze the recreational service provided by urban parks for local residents, we excluded two types of parks: small parks that had less than 10 check-in comments on the Dazhong Dianping platform (see Section 3.2.) because their feedback information is insufficient, and 5A and 4A level parks that mainly refer to the famous historic site parks, because they attract many non-local tourists each year. Finally, 90 ordinary parks were selected to conduct analysis in this study (Figure 2). The area of the selected parks ranged from 1.65 ha to 457.14 ha, with the mean size of parks of 44.92 ha.

### 3.2. Data Source

#### 3.2.1. Dazhong Dianping Check-in Comments on Parks

This study used social media check-in data to quantify the urban park vitality. Dazhong Dianping is a tried-and-true review app where Chinese locals rate local restaurants, entertainment, scenic spots, and parks and share their reviews, such as user experience, satisfaction, or dissatisfaction (http://www.dianping.com/). The check-in comments from Dazhong Dianping are open data uploaded by users. Each comment includes the user’s name, location, money spent, comment text, and satisfaction score for the park visited. There are two substantial advantages of using the check-in comments data: (1) They can effectively reveal the park visits, recreational services content, as well as the satisfaction rate; (2) the data are relatively easy to obtain through corresponding application program interfaces (APIs) without encountering privacy issues or data qualifications. Although the check-in data on Dianping cannot cover all parks, and several small parks have very limited comments, it has been proven to be useful in studying urban public activities as one of the most popular types of geo-tagged data in a city [43,44]. Using the intensity of check-in comments on parks (i.e., the number of check-in comments per hectare) as a proxy to reflect the visiting intensity of urban parks, combined with the recreational satisfaction rate from Dianping, make the urban park vitality measurable and easy to be visualized.

#### 3.2.2. POI Data

The POIs are specific point data of spatial entities closely related to our everyday lives with geographic information, such as longitude and latitude, as well as attribute information, including name, type [27,42]. They provide precise locations and detailed category information on commercial sites, living services, and public sites, such as residences, administration offices, subway stations, bus stations, schools, and parks, with various advantages including large quantities and free access. Furthermore, people’s social activities in public spaces can be represented by their interactions with POIs rather than by land-use type because the statistical granularity of POI data is much finer compared with land-use data [27]. Therefore, POI data are conducive to the measurement of urban function characteristics at finer scales [45], and also provide an approach to quantify the effects of urban function characteristics on urban green space.

The POI data in this study were obtained from the Gaode Map website (https://lbs.amap.com/), which is one of the largest Chinese electronic map navigation search engines. According to the *Code for classification of urban and rural land use and planning standards of development land* in China (GB 50137) [46] and related research [47], the initial twenty POI types were filtered and aggregated into seven more general functional categories (Table 1). After reclassification, there were 759,722 POIs in total in Beijing: commercial service POIs accounted for 47.61%, followed by public administration and service (18.44%), office building/space (17.54%), transport facilities (10.84%), residential communities (2.47%), education (2.22%), and parks and scenic spots (0.88%).

### 3.3. Measuring the Park Vitality

Visiting intensity is an important component of park vitality. On the other hand, the measurement of park vitality should also capture the satisfaction level of using recreational services from beneficiaries. High satisfaction means the high fulfillment level of recreational service quality, which is conceived as the degree to which environmental opportunities meet people’s preferences [31]. The freely available check-in comments on different parks from Dianping provide an approach for measuring park vitality. To a certain extent, the number of check-in comments per hectare in a park can reflect the visiting intensity that the park attracts. To ensure that the value of park vitality calculated based on the visiting density conforms to the normal distribution, we log-transformed the park visiting intensity because the check-in data were not normally distributed. The park vitality is calculated as:*Vitality* = *Satisfaction* × ln (*Intensity* + 1)(1)
*Intensity* = *Number*/*Size*(2)

For one park, *Vitality* is the park’s vitality value; *Satisfaction* was obtained from the check-in comments data at the park, with a value between 0–1; *Intensity* is the visiting intensity for the park; *Number* is the number of visits to the park, which uses the total number of check-in comments; *Size* is the park area (hm^2^). The *Number* divided by the *Size* is the visiting intensity for the park. To ensure that the park visiting intensity value after taking the logarithm is greater than 0, the visiting intensity value is added to by 1.

### 3.4. Determination and Calculation of Potential Impact Factors

From the cascade process of “recreational service produce–utilization–human well-being” [29], we selected the potential factors affecting park vitality from three aspects: internal supply factors (i.e., landscape elements and recreational facilities), external demand factors (i.e., population), and the spatial links in SCAs (i.e., traffic accessibility and urban function characteristics). We collected and calculated these potential factors based on POIs, population distribution, and other multi-source geographic datasets in ArcGIS 10.3 (Table 2).

#### 3.4.1. Supply Factors

When park visitors enjoy the recreational services, they are actually appreciating a mix of biotic, abiotic, and man-made park infrastructure elements and qualities [48]. Some studies have found that the ‘natural’ structures and ‘water features’ contribute the most to high park use and ratings of urban green spaces [49]. In addition to natural and water elements, recreational facilities (such as footpaths, sport facilities, toilets, picnic areas, food services, and stores) are important for encouraging physical activity in parks [20,48]. Previous research has indicated that urban parks closer to the city center tended to attract more visits [36]. In comparison, the green space resources in the urban center are scarcer than those in the suburbs [50]. Thus, we assume that spatial differences in the recreation supply are likely to affect the way people value the service benefits. Finally, referring to previous studies, we selected five potential supply factors, including the presence of an entrance fee, the presence of water, the percentage of vegetation cover, the density of facilities (proxied by the POI density of the park) in the park, and the distance from the park to the urban center.

#### 3.4.2. Demand Factors

The landscape and natural spaces in parks are inanimate. The existence of recreationists and their interaction with the natural space can manifest the vitality of parks [11,20]. The characteristics determining demand for recreational services are referred to as socio–demographic and socio–economic characteristics of the population. Due to that, the benefits of urban parks are mainly co-produced by the spatially accessible interactivity between residents and urban parks, and the population density around parks may play an important role because people who live closer to a park are more likely to visit it [51]. We choose the nearby average population density as the potential demand factor.

#### 3.4.3. Spatial Links

There are two main aspects of spatial links in the SCAs: traffic accessibility and urban function characteristics. Traffic accessibility determines the opportunities from the beneficiary’s area (i.e., residential area) to the recreational service supply area (i.e., parks) [18,20]. We selected the density of nearby bus stops to reflect the traffic accessibility. Some studies have proven that reasonable planning configurations and the complex diversity of land-use types can prolong activity intensities and improve the urban vibrancy at a city scale [27,32]. For example, high density and mixed land use are positively correlated with neighborhood vibrancy [42]. Similarly, the urban function diversity and density in the SCAs may also influence the vitality of nearby parks. Mixed urban function indicates the diversity of commercial facilities, public services, residential housing, and office buildings that provide multiple near-home destinations [52]. It is assumed that more mixed urban function would encourage the continuous presence of people in streets and public spaces, offer a variety of social activities and cultural exchanges [26,41], and thus contribute to increasing perceived accessibility (subjectively measured accessibility) to nearby parks and their willingness to visit parks. The POI datasets were used to construct a POI-based urban function mix/diversity and function density as a measurement of the fine-scale urban function characteristics [45]. Our measure of the POI-based urban function mix applied Shannon entropy [53,54] and was calculated as Equation (3). The POI-based urban functional density was obtained by dividing the total number of all POIs in the buffer zone of 1000 m around the park by the buffer zone area.
(3)$$MIX=-\overset{n}{\underset{i=1}{\sum }}P_{i}\;\ln P_{i}$$
where *P_i_* is the proportion of the POI-based urban function of *i* type in the 1000 m buffer zone around the park. As previously mentioned, there are seven types of POI-based urban functions. A higher *MIX* value suggests a higher level of diversity in the POI-based urban function and vice versa. The number of different POIs in the 1000 m buffer area of the park were counted according to the POI category by the Spatial Join Tool in ArcGIS 10.3.

### 3.5. Statistical Analysis

The descriptive statistics, nonparametric Kruskal–Wallis test, and multiple regression analysis were used to study the park vitality in SPSS 24.0. We used the Kruskal–Wallis test to compare differences in park vitality among different spatial locations (Beijing’s 2nd to 6th ring road along the urban–rural gradient), and a Bonferroni test for pairwise multiple comparisons. Simultaneously, we also compared the differences in the value of each vitality-measuring index among different spatial locations. The park visits, visiting intensity (proxied by the number and density of check-in comments on the park), and recreational satisfaction rate among different spatial locations along the urban–rural gradient were analyzed. Multiple regression analysis was used to investigate the impacts of potential factors on park vitality; the variance inflation factor (VIF) for each predictor variable and the mean VIF were calculated to test the multicollinearity among predictor variables.

## 4. Results

### 4.1. Spatial Heterogeneity of Park Vitality

The spatial distribution characteristics of park vitality and the value of each index measuring vitality in Beijing are shown in Figure 3, and their descriptions are shown in Table 3. The value of park vitality ranged from 0.36 to 4.46, with a mean value of 2.04 and a standard deviation of 1.15 (Table 3). The spatial pattern of the vitality of urban parks showed considerable variations (Figure 3D). According to the ring location along the urban–rural gradient, the average values of the park vitality in the corresponding regions were 3.46, 3.11, 2.41, 1.61, and 1.56 from the 2nd ring to the 6th ring, respectively. The hypothesis of an equal median vitality across different park locations was rejected (Kruskal–Wallis test, *p* < 0.001), which showed that the park vitality decreased significantly along the urban–rural gradient. The Bonferroni test revealed that the park vitality within the 2nd ring was significantly higher than those in the 4th-5th ring and 5th-6th ring area (*p* < 0.001). Meanwhile, the park vitality in the 2nd-3rd ring were also significantly higher than those in the 4th-5th ring and 5th-6th ring area (*p* < 0.001). However, there was no significant difference between the park vitality with the 2nd ring and 2nd-3rd ring, and the difference among the parks in the 3rd-4th ring, 4th-5th ring and 5th-6th ring was also not statistically significant. On the whole, the park vitality within the 3rd ring was significantly higher than that outside the 4th ring.

The values of each index of measuring park vitality also showed different spatial distribution characteristics. A total number of 41,684 check-in comments to the non-famous urban parks in the sixth ring road of Beijing were reported in the Dianping platform before July 2019, with an average of 463 check-in comments per park, a standard deviation of 658, and a range from 15 to 3561 (Figure 3A, Table 3). Of the 90 parks, 7 parks had more than 1500 check-in comments, and 28 parks had less than 100 check-in comments. The results of the Kruskal–Wallis test showed that the hypothesis of an equal median number of check-in comments among the 2nd to the 6th ring locations was accepted (*p* = 0.108); thus, the number of check-in comments in parks has no significant spatial variations along the urban–rural gradient.

The intensity of check-in comments on parks varied greatly by location (Figure 3B) The intensity of the check-in comments tended to be high in small parks within the 3rd ring road (parks filled by red and orange) and low for large parks outside the 3rd ring road (parks filled by blue and green). The check-in comments in parks per hectare ranged from 0.72 to 257.92, with a mean of 30.55 counts/ha and a standard deviation of 44.47. The hypothesis of an equal median intensity of the check-in comments across the 2nd to the 6th different regions was rejected (Kruskal–Wallis test, *p* < 0.001), which showed that the intensity of the check-in comments decreased significantly along the urban–rural gradient.

The spatial distribution of park satisfaction was relatively even (Figure 3C). The average satisfaction across 90 parks was 0.78, with a standard deviation of 0.122 and a range from 0.33 to 0.96. There was no significant difference in the recreational satisfaction level along the urban–rural gradient (Kruskal–Wallis test, *p* = 0.741).

### 4.2. Effects of Different Factors on Park Vitality

The results of the regression analysis (Table 4) showed that the mean VIF value was 1.690, with the VIF values for all the predictor variables being less than 3. The low individual and average VIF values indicated that multicollinearity is not a problem in this model. The results showed that six predictor variables, including two supply factors (the presence of water and the facility density in the park), one demand factor (nearby population density), and three spatial linking factors (the nearby density of bus stops, the POI-based urban function mix, and density), significantly affected the park vitality. Approximately 68.3% of the variation in park vitality was explained by the model. Parks with water present have higher vitality scores, and the adequacy of facilities in the parks is another significant factor promoting the vitality of the park. The nearby population density and the density of bus stops positively correlated with the park vitality. The standardized regression coefficients of the external POI-based mixed function and POI density in the SCAs were 0.236 and 0.322, and they were positively correlated with the park vitality at significance levels of 0.001 and 0.01, respectively. The presence of an admission fee, vegetation coverage and distance to the city center had no significant effect on the park vitality. Therefore, under the same conditions of park physical attributes, nearby population density, and public transportation, we firmly believe that the high density and mixed-function development of commercial facilities, residential housing, schools, and office buildings that provide multiple near-park destinations could contribute to park vibrancy.

## 5. Discussion

### 5.1. Advantages of Introducing Vitality to Measure the Intangible Benefits of Urban Parks

Due to the intangible benefits of urban parks mainly being co-produced by the spatial interactivity between residents and urban parks, this study introduced vitality to reflect these intangible benefits. The park vitality mainly depends on actual contact with parks by individuals, the utilization of different recreational activities, and the satisfaction of using the parks. We constructed a straightforward and easily operated approach to assess the park vitality based on visiting intensity and recreational satisfaction rates. This is a comprehensive evaluation method including an objective index and subjective satisfaction perception. Park vitality can more intuitively capture the contributions of intangible benefits to human well-being and be more easily understood and employed by policymakers for green space management compared with monetized evaluation.

Additionally, measuring the park vitality and exploring its influencing factors based on freely available Dianping check-in comments, POIs, and other multi-source data is a spatially transferable and highly feasible approach that could yield valuable information to facilitate the planning and management of urban green spaces for other megacities. Taking Beijing as an example, the evaluation results showed that the vitality of urban parks decreased along the urban–rural gradient. The less vibrant parks were mainly distributed in the southern area between the 4th–5th ring and the area outside the 5th ring (Figure 3D), which is mainly characterized by the developing transitional urban–suburban zone. Compared with Figure 3B,C, we can see that the low vitality of these parks was mainly caused by the low visit intensity at the parks because the recreational satisfaction was evenly distributed overall, suggesting that the utilization of urban green space in suburban areas should be improved. In these urban fringe areas, large country parks or forest parks account for a large proportion, in which the natural landscape elements are relatively rich. The vitality of the park may be promoted by improving the public transportation convenience or the urban function diversity around the park as well as focusing more attention on the construction of man-made recreational infrastructures in the large suburb parks. Information from social media and POI data makes it highly efficient to evaluate the vitality of urban green spaces and study how people interact with nature-based environments at city-wide or block scales.

### 5.2. Recreational Service Supply and Demand Factors and Spatial Links Influencing the Park Vitality

We explored the factors influencing park vitality from the perspectives of recreational service supply, demand, and spatial linking characteristics. Compared with other studies focusing only on the internal park attributes and external socioeconomic environment, this research highlights the significantly important role of surrounding POI-based urban function mix in park vitality at the urban block scale. From the recreational supply factors, the presence of water and facility density in the park were positively related to the park vitality, a result which concurs with other relevant studies. Nordh (2013) found that proximity to water is highly valued in small urban green spaces [49]. Parks with water may have better aesthetic views and recreational value (such as providing esthetic appreciation or activities such as boating), and the parks with more well-maintained facilities, such as sport fields/courts, may have greater potential to increase exercise or recreational activities [21], providing higher recreational utilization and satisfaction compared to parks with a low density of recreational facilities [55]. Our study did not find a significant correlation between vegetation coverage and park vitality, which is in line with previous studies in Beijing. Zhang (2018) also found that the vegetation cover rate had no significant influence on park visits [36]. This may be because we did not consider landscape quality factors, such as the diversity of vegetation species. Distance to the city center was not a significant factor affecting the park vitality in multiple linear regression analysis when controlling for the effects of other independent variables, although the previous Kruskal–Wallis test showed there were significant differences in park vitality among different spatial ring locations along the urban–rural gradient. This suggests that the correlation between the spatial location and park vitality may be due to the correlations between the spatial location and population density or urban function variables. The nearby population density was positively correlated with park vitality, which means that parks in densely populated areas can be more frequently visited. Similar to previous studies, the green and blue areas may be perceived as more important to the quality of life by individuals in densely populated areas, because of the additional value of scarce natural resources [23]. In addition, due to the convenience of access to the park, nearby dense residents would account for a higher percentage of park visitations [56]. Thus, recreation potential could be effectively and fully used to generate recreation service benefits. The external density of nearby bus stops was also found to have a significant positive influence on park vitality because convenient accessibility via public transportation leads to higher park use [36,56], thereby increasing park vitality. This finding is consistent with earlier studies showing that active transport to recreation sites was strongly associated with frequent use by the youth [55], and it supports the need for transport policies that facilitate public transport to increase park vitality in Beijing [36].

A positive association between the POI-based urban function mix and density and park vitality was found in our study. This is similar to the results of a previous study [57], in which residents from areas with a higher land-use mix index were more likely to report active park use in Bogotá, Colombia. In high-density and mixed land-use urban environments, living in an area combined with various active destinations could motivate people to leave their houses and go for a walk, provide more opportunities for communication with others due to residents’ diverse experiences in daily life [42], and may result in an urban form that encourages interaction between the neighborhood and nearby park use [57,58]. Some studies have indicated, however, that higher neighborhood land-use mix did not contribute to the likelihood of an adjacent park being used [59]. This may be related to the socioeconomic status of the surrounding population. There are no available statistical data on information such as the occupation and income of the population at the block scale in China, and the census data from the administrative districts are not suitable for analyzing the 1 km buffer zone of the park; therefore, the confirmation of factors related to socioeconomic status, such as income and occupation, can only be obtained through time-consuming ground survey data and further analysis.

### 5.3. Implications for Urban Green Space Planning

To ensure the delivery of urban ecosystem services, we need heterogeneous, multifunctional, and accessible urban green space throughout our cities; however, “nature in cities” is bound to compete with other land uses and infrastructure for resources and space [5], and developing suburban areas or new towns with rapidly growing populations may face trade-offs between the growing demand for built-up infrastructure and green space [24]. Our study highlights the significantly important role of high POI-based mixed urban function in adjacent park vitality, which can contribute to urban green space design and management.

High density, good public transportation accessibility, and mixed land-use development are the main parameters of compact cities [60]. The mixing of development land uses as part of the urban Smart Growth movement has become one of the key principles of Chinese land-use planning and sustainable development strategies in recent years [42,61]. A balanced mix of working, service, schools, transport facilities, and living activities that provide multiple interconnected, active destinations could promote urban vibrancy and yield socioeconomic benefits (i.e., reducing fuel consumption for transportation) [62]. Our research also confirmed that POI-based mixed and high-density urban functions could promote adjacent park vitality at the urban block scale. Greater mixing of urban function facilitates walking, gathering, and other diverse activities, which could increase opportunities to coordinate walks in the nearby parks among locals as well as enhance the synergy between urban green space benefits and the economic benefits of the surrounding development land.

In the urban cores with high population densities and compact land functions, it is extremely unlikely that additional large parks will be established, due to the high opportunity costs from foregone property development [31]; however, small parks with easy access could be built in urban cores [63]. Meanwhile, in the developing suburban areas or new town areas, improving the urban functional density and diversity as well as the traffic accessibility of the parks’ peripheral areas, or increasing more man-made recreational infrastructures in parks, could achieve better urban park vitality and thus improve their benefits to human well-being.

### 5.4. Limitations and Future Research

This paper used the intensity of check-in comments at parks (i.e., the number of check-in comments per hectare) from Dianping as a proxy to measure the visiting intensity of urban parks, which is one of the important indicators of assessing park vitality in this case study. This proxy may be more time-efficient and can provide good spatial coverage; however, it may not be representative of all park users, since the elderly and children rarely use the open network platform to comment after visiting the park. Second, there may be a lack of comment data for a few small community parks, so the results of park vitality in this paper may have some deviation from the overall actual park utilization. This potential data bias may contribute to uncertainty in the analysis of the effects of influencing factors [36]. The reliability and usefulness of the Dianping check-in comments data will be further validated when visitor statistics are publicly released by park authorities in the future. It is possible that park visiting intensity based on mobile phone signaling data would be more comprehensive and accurate [56], but this type of data lacks information on visitors’ satisfaction when using the parks. Nevertheless, we can still be sure that the results of this study are valuable in revealing the characteristics of park benefits and providing a scientific basis for optimizing green space construction.

Finally, this paper does not analyze in-depth the impact of factors related to the quality of recreational services supply (such as park cleanliness, safety, and the diversity of biological and abiotic features) on park vitality. This can be achieved through semantic analysis of the text content of the check-in comments at parks. For example, detailed factors related to the subjective perception and landscape sense of recreation can be deeply explored from the aspects of visual aesthetics, smells, natural sounds, and tactile experiences in urban green spaces [64], which warrants further research. Furthermore, at different spatial scales (such as at street, block, or city scales), a comparative study analyzing the effects of the POI-based urban function characteristics on park vitality for different types of cities may be interesting and informative.

## 6. Conclusions

From the perspective of the spatial interactivity between residents and urban parks, this paper introduced “vitality” to understand and measure the “intangible” benefits of urban parks. Taking advantage of Dianping check-in comments, POIs, and other multi-source datasets, this study quantified and mapped the urban park vitality in Beijing, and then investigated factors that affect urban park vitality from three dimensions involving recreational service supply, demand, and spatial linking characteristics.

Compared with conventional park use benefit-related studies, this study makes two innovative improvements. First, based on “recreational service produce–utilization–human well-being” cascade view, we present a clear approach for assessing urban park vitality to measure the intangible benefits of using urban parks, which make the intangible benefits easily measurable and straightforward for urban green space management and optimal configuration at a finer spatial granularity. Moreover, this way of assessing the benefits avoids the biases caused by assigning a virtual price and some relatively subjective measurements in questionnaires. The second innovative improvement is that the effects of spatially explicit urban function characteristics (i.e., the POI-based urban function mix and density) on adjacent park vitality were emphasized and explored, which could provide management implications for enhancing the vitality of urban green spaces from the aspect of optimizing urban form and structure.

The results of this study showed that, from the supply side, the presence of water and facility density in the park have significant impacts on urban park vitality; the demand factors, such as the surrounding population density, were also positively related to park vitality. Moreover, higher levels of POI-based urban function mix, function density, and public transportation-friendly streets were associated with greater vitality of adjacent parks. Compared with other studies focusing only on the internal park attributes and external environment, the findings of our study shed light on park vitality in relation to the urban function characteristics at urban micro or block scales, which suggests that infilling small parks in urban core areas where the urban function is compact while moderately increasing the land functional density and diversity of the park peripheral area in urban developing districts or urban–rural areas could achieve better urban park vitality and thus improve human well-being. The vitality of urban parks, which represents the intangible benefits of using urban parks, may serve as a useful and feasible gateway for addressing and managing nature in cities.

## Figures and Tables

**Figure 1 ijerph-17-01615-f001:**
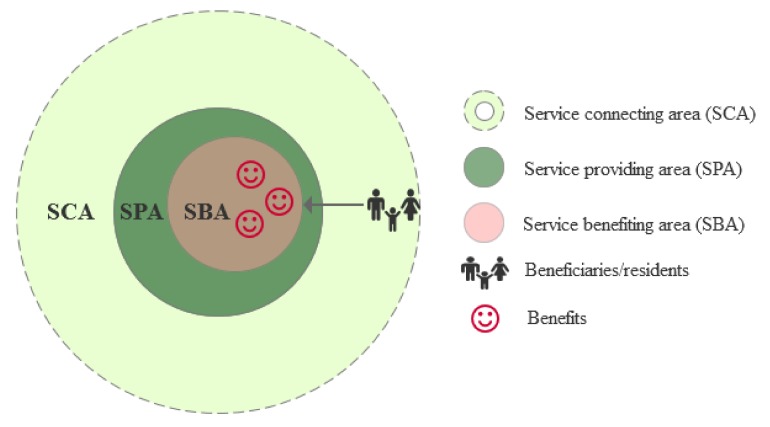
Spatial relationship between the recreational service providing area (SPA) and service benefiting area (SBA) (own draft based on Syrbe (2012), Syrbe (2017)) [17,19]). Recreational services in urban parks are user movement-related ecosystem services, where SBA is equal or similar to SPA because people must be in the SPA to benefit from the recreational services. The light green area is the service connecting area (SCA), such as the surrounding area of the urban park and nearby roads, which connects beneficiaries (people in residential or office areas) to the SPA.

**Figure 2 ijerph-17-01615-f002:**
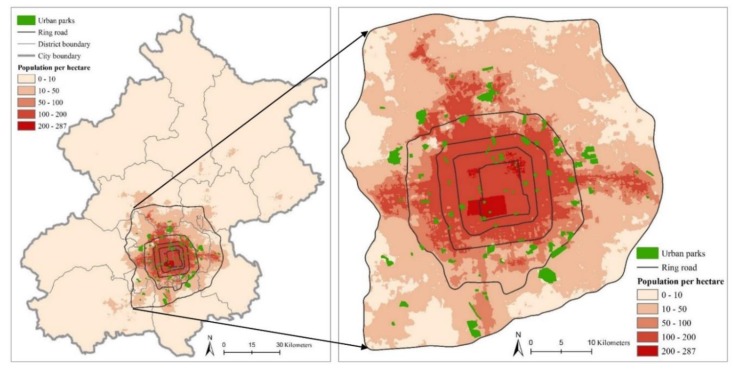
The spatial distribution of the urban parks and population density in the study area, from the inside to the outside, are the 2nd, 3rd, 4th, 5th, and 6th ring roads along the urban–rural gradient.

**Figure 3 ijerph-17-01615-f003:**
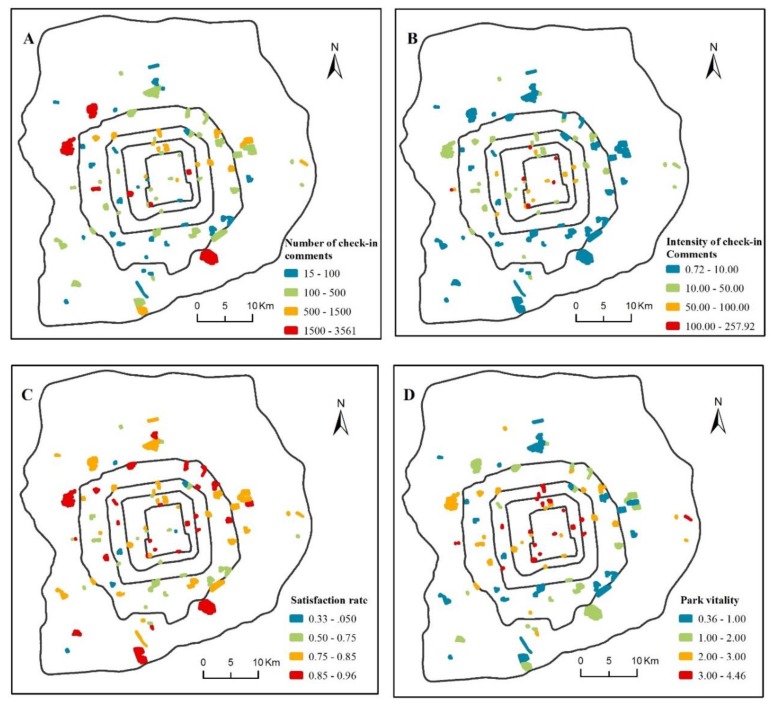
Spatial distribution of the total number of checked-in comments (**A**), the intensity of checked-in comments (**B**), recreational satisfaction rate (**C**), and the vitality (**D**) of urban parks in the study area.

**Table 1 ijerph-17-01615-t001:** Number of points-of-interest (POIs) in each urban functional category in Beijing.

POI-Based Urban Function Type	Counts	Proportion
Residential communities	18,730	2.47%
Public administration and service	140,115	18.44%
Education	16,852	2.22%
Commercial service	361,722	47.61%
Office building/space	133,291	17.54%
Transport facilities	82,336	10.84%
Parks and scenic spots	6676	0.88%

**Table 2 ijerph-17-01615-t002:** Descriptions and data sources of selected potential factors affecting park vitality.

Type	Variables	Definition	Measurement	Data Sources/Description
**Supply factors**	Fee	Entrance fee of park or not	1 = yes, 0= not	Beijing Municipal Bureau of Landscaping
	Veg_Rate	Vegetation cover rate in park	%	Based on the natural related vector data in 2015 from Beijing GISUNI Information Technology Corporation
	Water	Presence of water in park	1 = yes, 0 = not	Based on the natural related vector data in 2015 from Beijing GISUNI Information Technology Corporation
	Fac_Den	The density of facilities (proxied by POI density) in park	Count/ha	Based on the POIs data from Gaode Map in 2018, calculated in each park.
	Dis_center	Distance from park centroids to urban center	Kilometer	Calculated in ArcGIS 10.3
**Demand factors**	Pop_Den	Population density outside the park	Population/ha	Provided by United Nations Statistics Division (https://unstats.un.org/home/), raster dater with 100 m resolution in 2015, calculated within the 1000-m-buffer zone
**Spatial link** **factors**	Stop_Den	Density of bus stops outside the park	Count/ha	Provided by Gaode Map in 2018, calculated within the 500-m-buffer zone
	Func_Mix	The POI-based urban function mix/diversity outside the park	≥0	Provided by Gaode Map in 2018, calculated within the 1000-m-buffer zone
	Func_Den	The POI-based urban function density outside the park	Count/ha	Provided by Gaode Map in 2018, calculated within the 1000-m-buffer zone

**Table 3 ijerph-17-01615-t003:** Summary statistics of check-in comments, recreational satisfaction rate, and vitality of parks (N = 90).

Catagory	Minimum	Maximum	Mean	Std. Deviation
Number of check-in comments on parks	15.00	3561.00	463.16	658.56
Intensity of check-in comments on parks	0.72	257.92	30.55	44.47
Satisfaction rate on parks	0.33	0.96	0.78	0.12
Park vitality	0.36	4.46	2.04	1.15

**Table 4 ijerph-17-01615-t004:** From multiple linear regression analyses of park vitality on different factors.

Table.	Variable	Coefficient	St_Coefficient	*p*	VIF
**Supply factors**	Fee	0.322	0.088	0.194	1.147
	Veg_Rate	0.267	0.076	0.321	1.455
	Water	0.505 **	0.206	0.008	1.443
	Fac_Den	0.074 *	0.186	0.015	1.416
	Dis_center	−0.003	−0.018	0.847	2.107
**Demand factors**	Pop_Den	0.002 *	0.165	0.045	1.659
**Spatial link factors**	Stop_Den	0.848 *	0.183	0.029	1.713
	Func_Mix	1.860 ***	0.236	0.001	1.300
	Func_Den	0.086 **	0.322	0.004	2.970
	R^2^	0.683			
	Adjusted R^2^	0.647			
	Mean VIF	1.690			

*, **, *** indicate significance at the 0.05, 0.01, and 0.001 levels, respectively.

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
