# Peer review of "Vitality of Urban Parks and Its Influencing Factors from the Perspective of Recreational Service Supply, Demand, and Spatial Links"

_ijerph, 2020, doi:10.3390/ijerph17051615_

Round 1
Reviewer 1 Report
The strong side of the publication is the simplicity of the idea, and thus the ease and speed of evaluation. The introduction of the vitality index by the authors is a way to assess complex phenomena and therefore has a chance to be easily applied in spatial planning. The publication is worth publishing. The authors base their conclusions on the vitality index, which is strongly generalized and not devoid of flaws, they have made frequent simplifications in the interpretation of the results (which I described below). In particular, in a short chapter "5.4. Limitations and future research, the authors have silenced the imperfections of source data.
Notes on the "satisfaction" indicator. Assessments of park users "satisfaction" in highly urbanized cities like Beijing as well as European or American cities seem to be flattened. This is usually achieved by ratings 4.3-4.5. This is due to the fact that greenery is treated as something related to other city facilities, which are located in built-up areas. The "satisfaction rate" rating as an indicator 0-1 ranks parks in relation to the general opinion about parks, but it is not a, in fact, the rating of peoples well-being. The concept of well-being despite different approaches (e.g. Linton M.J., Dieppe P., Medina-Lara A. 2015. Review of 99 self-report measures for assessing well-being in adults: exploring dimensions of well-being and developments over time. BMJ Journals) has a clear health reference that is not reflected in the ratings on google.map or Gaode.map. Users often assess their impressions (rare events) and lesser or even unconscious health and wellbeing impacts. The "satisfaction" indicator, therefore, refers to the attractiveness of a place and has little to do with ecosystem services, of which there are many mentioned in the article and in particular in the chapter "5.3. Implications for urban green space planning".
Comments on the title indicator 'vitality', referring to the intangible benefits of urban parks. The indicator strongly depends on the number of ratings and the size of the park (facilities in figure 3 that achieve the highest satisfaction rate often have low vitality rates). The number of ratings does not necessarily reflect the time spent in the park and the ecosystem services resources. It is more common for visitors who have been to the facility for the first time to leave opinions online than for residents who spend time there on a daily basis. This indicator is correlated with accessibility, which is shown by the accessibility indicator for buses (Stop-Den, Table 2), or strictly urban zones in relation to suburbans. The "vitality" indicator is, therefore, a strongly generalized indicator, showing not so much "intangible assets from urban parks" as their attractiveness and, to a greater extent, their accessibility. The obtained results and min. maps of figure 3 are of practical value, but one should take into account the limitations in their interpretation. Please eliminate these generalizations.
Factors influencing the vitality index. According to table 2 the presence of water and availability of buses are the most important factors, which is justified in the literature. However, a lot of space has been devoted to the calculation of the MIX indicator (it should be noted that MIX value is an indicator of Shannon's diversity) presenting the role of various forms of land use according to POI. It seems to me that places of concentration of people, meeting places with a large number of people have a structure similar to that of diversity. Please develop the discussion and explain what this mixed land use around parks would be about. Isn't this a secondary effect of attractive parks? A known fact is the increase in the value of land around green areas and ecogentrification of space. Complete the discussion.
The lack of connection with vegetation cover has been shown, although the authors indicate in the discussion that large trees seem to be important. Therefore I suggest to include tree cover in the analyses. The most accurate solution would be to use the NDVI remote sensing indicator available in the literature for Beijing in the park (Zhang X.X., Wu P.F., Chen B. 2010. Relationship between vegetation greenness and urban heat island effect in Beijing City of China. Procedia Environmental Sciences 2, 1438-1450).
Author Response
Thanks very much for your kind comments concerning our manuscript. Those comments are all valuable and very helpful for revising and improving our paper, we have learned a lot from it. After seriously studying your comments and advice, we have made corresponding changes in the updated document.
Our responses to the comments are in the attachment.

Reviewer 2 Report
This paper constructs an index to attempt to measure cultural ecosystem service production/provision from parks. Such work is important as cultural ecosystem service receive relatively little treatment in comparison with other categories of ecosystem service due to the difficulty of measuring them. To my mind it is especially valuable that this paper attempts to quantify ecosystem services without using an economic metric, and using widely-available data sources. Unfortunately, in its present state there are a number of flaws, most concerningly around the conceptual framework that places visiting intensity and satisfaction of recreational demand as proxies for cultural ecosystem service provision (without, in my view, adequate justification). In general, the work is also not adequately placed in the context of existing research in the field.
I have not recommended rejection because I do believe that the aims of the work are important. However, I think substantial additional work is required before it is suitable for publication, because I think that the concept of park ‘vitality’ (i.e. delivery of intangible benefits via cultural ecosystem services) is not appropriately constructed or adequately robust at this point.
Introduction
There is very little in the way of literature review. This section should be expanded in order to place the present study in context of work that has already been done to understand how cultural ecosystem services are produced and provided. Page 2, lines 50-55: The description of service providing and service benefiting areas is inadequate. It is not clear how the two things differ. Page 3, lines 80-104, but also more generally throughout the paper: There is no explanation of why you think vitality/visiting intensity is an adequate proxy for cultural ecosystem service provision. Places that are visited regularly may simply be more conveniently situated, and not provide much benefit to well-being. In contrast, less easily accessible sites may provide far greater benefits to the people who do visit. Similarly, there is no justification for why you think recreational satisfaction as measured from a review (which, notably, was not reviewing perceived benefits to well-being) is an accurate reflection of actual, objective benefits. I think this is a serious flaw that undermines the conceptual framework of the paper. Page 3, lines 121-129: It is unclear what an “urban functional structure” is.
Methods
Page 4, lines 151-152: You exclude certain park types due to high rates of visitation by tourists. How many tourists visit other parks? Is the presence of tourists biasing your results? It seems this is possible given that people may be more likely to ‘check-in’ to places not in their usual neighbourhood, or review places that are not part of their everyday experience. Page 4, lines 151,154: Is this all the parks not excluded due to parks with insufficient check-ins and 5A/4A parks? If not, on what basis did you select them? Some of the parks charge an admission fee – did this bias results by excluding certain residents? Page 6, lines 206-213: The form of the equations needs justification – why did you take the logarithm of intensity? Why do you think the relationship between number of visitors and size should be linear? This is important given the result that intensity varies with distance from urban centre. Page 6, lines 226-232: More justification is needed for your choice of factors, and you should also describe what relationships you would expect to see. E.g. percent vegetation cover may be relatively unimportant compared to the types of vegetation present, either in the form of the presence of individual vegetation types, or the diversity/complexity of the landscape. Both of these are known to influence human responses to greenspace as well as biodiversity (which in turn can influence human responses). Additionally, it is unclear why distance from the park to the urban centre is a supply factor. If population densities are higher in the urban centre then surely this is confounded with demand. In general, I think the supply factors include those effecting both to how much benefit can be provided per person, and how many people the benefits will be accessible to. It may be helpful to separate these out, or at least make them explicit. Page 6, lines 235-239: Why is house price a demand factor? What relationship are you expecting to see between house price and demand for cultural ecosystem services, and why? Again – justifications and hypotheses are needed for these factors (and the spatial link factors).
Results
Page 8, lines 282-283: You did not report how you identified which groups were significantly different (i.e. which post-hoc tests you used). Page 10, lines 314-314: It does not follow from the lack of multicollinearity that the residuals are unbiased.
Discussion
Page 11, line 354: How should the utilisation of greenspace in urban areas be improved? It seems a natural consequence of cities designed around urban centres that central amenities will be more crowded. Page 11, lines 360-383: I would like to see much more reference to other literature in this paragraph so that your results can be placed in the context of existing research, and the reasons for the patterns you observed can be better understood. Page 11, line 378-380: It could also just mean that people are more likely to check-in/leave reviews for places near the urban centre. Also – do central urban greenspaces provide more opportunities for connecting with nature? In my (decidedly Western European) experience, greenspaces near the urban centre provide relatively little opportunity for nature connection compared to suburban greenspaces, due to the dearth of biodiversity and naturalistic planting, though this may be different elsewhere. Page 12, lines 408-411: There is recent literature suggesting that high-density cities are in fact not so sustainable and this should be recognised (or, alternatively, do not make the statement, given that it is not particularly relevant to your study). Page 12, line 422: What do you mean by “smaller filled parks”? Page 12, lines 440-442: I’m not sure that I agree with your statement that you “can be sure” that the study provides a scientific basis for “optimising green space construction”, both given the potential for bias in your data sources that you haven’t been able to explore properly (due to the nature of the platform), and all the limitations mentioned in my comments that haven’t been acknowledged.Author Response
Thanks very much for your kind comments concerning our manuscript. Those comments are all valuable and very helpful for revising and improving our paper, we have learned a lot from it. After seriously studying your comments and advice, we have made corresponding changes in the updated document.
Our responses to the comments are in the attachment.

Round 2
Reviewer 2 Report
The authors have carefully addressed each of my concerns and made significant improvements to the manuscript. I am now pleased to recommend the paper for acceptance. It is enjoyable to read and makes a valuable contribution to urban cultural ecosystem service research.